# A Tight Lower Bound and Efficient Reduction
# for Swap Regret

**Shinji Ito**
NEC Corporation
`i-shinji@nec.com`

## Abstract

Swap regret, a generic performance measure of online decision-making algorithms, plays an important role in the theory of repeated games, along with a close connection to correlated equilibria in strategic games. This paper shows an $\Omega(\sqrt{TN \log N})$-lower bound for swap regret, where $T$ and $N$ denote the numbers of time steps and available actions, respectively. Our lower bound is tight up to a constant, and resolves an open problem mentioned, e.g., in the book by Nisan et al. [28]. Besides, we present a computationally efficient reduction method that converts no-external-regret algorithms to no-swap-regret algorithms. This method can be applied not only to the full-information setting but also to the bandit setting and provides a better regret bound than previous results.

## 1 Introduction

Online decision-making has been extensively studied owing to its wide range of applications to online learning [32], repeated game [11; 21], and algorithmic game theory [28]. In this study, we consider an online decision-making problem with a finite set of actions $[N] = \{1, 2, \ldots, N\}$, where $N$ denotes the number of actions. In each time step $t \in \{1, 2, \ldots\}$, the environment chooses the loss vector $\ell^t = (\ell_1^t, \ldots, \ell_N^t)^\top \in [0, 1]^N$, while a player chooses an action $i^t \in [N]$ without knowledge of $\ell^t$, and incur a loss of $\ell_{i^t}^t$ for the chosen action. After choosing an action, the player gets feedback information regarding the loss vector $\ell^t$. Two different settings are pondered in terms of feedback. The *full-information* setting allows the player to observe all entries of $\ell^t$, whereas the *bandit* setting reveals only the loss $\ell_{i^t}^t$ for the chosen action. This study deals with the adaptive adversary model, i.e., $\ell^t$ may behave adversarially depending on the actions $i^1, \ldots, i^{t-1}$ chosen so far.

*External regret*, or simply *regret*, is a typical measure of the performance of online decision-making algorithms, which is defined to be the difference between cumulative losses for the algorithms and for the single best action in retrospect. Algorithms achieving external regret of sublinear order in $T$ are called *no-external-regret* algorithms, where $T$ denotes the total number of time steps. This no-external-regret property implies that the averaged performance of chosen actions converges to, or are superior to, that for any single action. It is duly recognized that expected external regrets of $O(\sqrt{T \log N})$ can be achieved in the full-information setting [4; 18], and of $O(\sqrt{TN})$ in the bandit setting [5].

*Swap regret* [9] is an alternative performance measure that compares the cumulative loss for the algorithm and that for *swapped* action sequences generated by arbitrary *modification rules* $F : [N] \to [N]$. To define swap regret, we produce a sequence of swapped actions $F(i^1), F(i^2), \ldots, F(i^T)$, and evaluate the difference between cumulative losses for these action sequences and the original sequence, which can be expressed as $R^T(F) = \sum_{t=1}^{T} (\ell_{i^t}^t - \ell_{F(i^t)}^t)$. Swap regret is defined as the maximum of $R^T(F)$ over all $N^N$ functions $F$ from $[N]$ to $[N]$. If an algorithm has a swap-regret bound of sublinear in $T$, it is termed a *no-swap-regret* algorithm. This property is stronger than the

Table 1: Bounds for swap regret. Lower bounds apply both to the full-information setting and to the bandit setting.

| Feedback | Upper bound, computational cost | Lower bound |
|----------|--------------------------------|-------------|
| Full-info. | $O(\sqrt{TN\log N})$, poly-time [9; 27] | $\Omega(\sqrt{TN})$ [9] |
| Bandit | $O(\sqrt{TN^3\log N})$, poly-time [9] $O(\sqrt{TN^2\log N})$, exp-time [33] $O(\sqrt{TN^2})$, poly-time (**Theorem 2**) | $\Omega(\sqrt{TN\log N})$ (**Theorem 1**) |

no-external-regret property, i.e., no-swap-regret algorithms are also no-external-regret algorithms. In fact, $F$ can be a function that maps all actions to a single action $i \in [N]$, which implies that swap regret is an upper bound of external regret. Whereas, the no-external-regret property does not guarantee the property of no-swap-regret [34]. A major application of no-swap-regret algorithms can be found in the field of the algorithmic game theory. For any strategic games, given a no-swap-regret algorithm, one can construct an algorithm for computing approximately correlated equilibria [16; 20; 22].

Blum and Mansour [9] and Stoltz [33] have provided no-swap-regret algorithms, both for the full-information setting and for the bandit setting, as can be seen in Table 1. The algorithms by Blum and Mansour [9] are based on reduction methods that, given no-external-regret algorithms $\mathcal{A}$, convert it to a no-swap-regret algorithm that calls $\mathcal{A}$ as subroutines. For the full-information setting, their approach achieves $O(\sqrt{TN\log N})$-swap regret, while they have shown that the swap regret is at least $\Omega(\sqrt{TN})$ in the worst case. The gap of $\sqrt{\log N}$-factor between the upper and lower bounds has remained, and removing this gap has been mentioned as an open problem, e.g., in Chapter 4 of the book by Nisan et al. [28] and in the thesis of Stoltz [33]. For bandit setting, Blum and Mansour [9] proposed an algorithm that achieves swap regret of $O(\sqrt{TN^3\log N})$, and Stoltz [33] achieved $O(\sqrt{TN^2\log N})$.

**Our contribution**

This study's contribution is two-fold: one is a tight lower bound for swap regret, and the other is a novel efficient method for achieving no-swap-regret. The first result can be summarized in the following theorem:

**Theorem 1.** *Suppose that $N \geq 2^{64}$ and $4N\log N \leq T \leq N^{3/2}/(2^8\log N)$. There exists an adaptive environment for which any randomized online algorithm suffers swap regret bounded as*

$$\mathbf{E}\left[\max_{F:[N]\to[N]} R^T(F)\right] \geq 2^{-11}\sqrt{TN\log N}.$$

The proof of this theorem is given in Section 4. This theorem shaves off the above-mentioned $O(\sqrt{\log N})$-gap between the upper and lower bounds for swap regret. The concluding analysis indicates that the minimax optimal bound for swap regret is $\Theta(\sqrt{TN\log N})$, and that there is no room for improving $O(\sqrt{TN\log N})$-swap-regret algorithms, except for constant factors.

*Remark.* The assumption of $T = \Omega(N\log N)$ in Theorem 1 is inevitable to prove an $\Omega(\sqrt{TN\log N})$-lower bound, as the swap regret is at most $O(T)$ from the definition. The other assumption of $T = O(N^{3/2}/\log N)$ can be relaxed to $T = O(N^{2-o(1)})$ via a minor modification to the proof. We should, however, note that the latter assumption is stronger than for the previous $\Omega(\sqrt{TN})$-lower bound [9], in which $T \leq \frac{1}{\sqrt{N}}\exp(N/288)$ is assumed. We conjecture that the assumption of $T = O(N^{3/2}/\log N)$ in Theorem 1 can be removed by a more sophisticated analysis.

To prove Theorem 1, we refine the analysis of the $\Omega(\sqrt{NT})$-lower bound given by Blum and Mansour [9]. They pondered an adaptive environment in which each loss follows a Bernoulli distribution of parameter $1/2$ independently for most actions and time steps. Here, it can be observed that under the assumption that the chosen actions are *balanced*, i.e., if there are $\Omega(N)$ actions each of which is chosen in $\Omega(T/N)$ time steps, the expected swap regret is at least $\Omega(N \cdot \sqrt{\frac{T}{N}\log N}) =$

$\Omega(\sqrt{TN\log N})$. This follows from the fact that the expectation of the loss $\ell_{i^t}^t$ is $1/2$ independent of the algorithm, and an extreme value analysis providing that the minimum of $N$ independent variables following binomial distribution $Bi(n, 1/2)$ is $n/2 - \Omega(\sqrt{n\log N})$ asymptotically (corresponding to $\min_{j\in[N]} \sum_{t\in[T]:i^t=i} \ell_j^t$, where $n = n_i := |\{t \in [T] : i^t = i\}|$ and $j = F(i)$). However, without the assumption of "balanced choice", e.g., if the algorithm chooses a single action in all time steps, then the derived lower bound is only $\Omega(\sqrt{T\log N})$. To ensure the algorithm's decision to be balanced, Blum and Mansour [9] modified the environment so that actions chosen $\Omega(T/N)$ times so far are *blocked*, i.e., returns the loss of 1. Then, it can be inferred that the chosen actions are balanced. Indeed, if the assumption does not hold the algorithm must choose blocked actions $\Omega(T)$ times, which causes $\Omega(T)$-swap regret. In this modified environment, however, the extreme value analysis used above cannot be applied as yet because the value of $\sum_{t\in[T]:i^t=i} \ell_j^t$ does not follow the binomial distribution if the $j$-th action is blocked. Noting that there can be $O(N)$ blocked actions, if actions with lowest losses are blocked, we have $\min_{j\in[N]} \sum_{t\in[T]:i^t=i} \ell_j^t = n_i/2 - \Omega(\sqrt{n_i})$, which provide $\Omega(\sqrt{TN})$-lower bound for swap regret as shown in [9].

In this present study, we improve upon the analysis to provide an $\Omega(\sqrt{TN\log N})$-lower bound, by showing that the probability that "good" actions are blocked for most actions $i$ is sufficiently small. More precisely, we show that $\min_{j\in[N]} \sum_{t\in[T]:i^t=i} \ell_j^t = n_i/2 - \Omega(\sqrt{n_i}\log(\sqrt{N}))$ holds for $\Omega(N)$ actions $i$ with a high probability, which is concluded from a combination of a property of order statistics, a union bound, and concentration inequalities.

The other contribution of this study can be summarized as follows:

**Theorem 2.** *Suppose that there exists an $r(T)$-external-regret (in expectation) algorithm $\mathcal{A}$ for $N$ actions, where we assume $r(T)$ to be a concave function in $T$. Suppose that the computational time taken to run $\mathcal{A}$ for $T$ time steps is bounded by $\mathsf{Init}_\mathcal{A} + T \cdot \mathsf{Step}_\mathcal{A}$.[1] Then, we can construct an algorithm $\mathcal{B}$ for which the expected swap regret is bounded by $N \cdot r(T/N)$. The total time complexity of $\mathcal{B}$ is bounded by $(N \cdot \mathsf{Init}_\mathcal{A} + T \cdot \mathsf{Step}_\mathcal{A} + O(T \cdot \mathsf{SD}_N))$, where $\mathsf{SD}_N$ stands for the time taken to compute a stationary distribution of a given Markov chain with $N$ states.[2]*

A constructive proof of this theorem is given in Section 5. We note that this theorem can be applied both to the full-information setting and to the bandit setting, i.e., if $\mathcal{A}$ works for the bandit setting, then so $\mathcal{B}$ is. Because there is an $O(\sqrt{NT})$-external-regret algorithm for the bandit problem [5], the above theorem implies that the expected swap regret of $O(N\sqrt{T})$ can be achieved. As can be seen in Table 1, the regret bound is better than the existing bound shown by Stoltz [33] and can be achieved by a computationally efficient algorithm. Besides, the reduction method here is more generic and efficient, compared to the ones proposed by Blum and Mansour [9]. More precisely, reduction methods given in [9; 27] require additional assumptions referred to as *data-dependent (first-order) bounds* for external-regret [1; 7; 12; 24], which is a refined bound depending on the cumulative loss (or reward) rather than the number of rounds. In contrast to these existing reduction methods, the proposed reduction method in Theorem 2 works without assumptions of data-dependent regret bound for $\mathcal{A}$. In addition, existing no-swap-regret algorithms presented in [9] require a computational time of $(N \cdot \mathsf{Init}_\mathcal{A} + NT \cdot \mathsf{Step}_\mathcal{A} + O(T \cdot \mathsf{SD}_N))$ in the notation of Theorem 2. On the other hand, our algorithm's time complexity regarding $\mathsf{Step}_\mathcal{A}$ is only $(T \cdot \mathsf{Step}_\mathcal{A})$, which improves upon the previous results with a factor in $N$. The algorithm proposed by Mohri and Yang [27] runs in $(N \cdot \mathsf{Init}_\mathcal{A} + NT \cdot \mathsf{Step}_\mathcal{A} + O(N^2 T \log T))$-time, which includes an $O(N^2 T \log T)$-term in place of $O(T \cdot \mathsf{SD}_N)$. This time complexity is incomparable to ours in general, though ours is superior under the condition of $\mathsf{Step}_\mathcal{A} = \Omega(N^{1+\Omega(1)})$.

Our reduction method for proving Theorem 2 is inspired by [9] as well. Similar to their reduction methods, our algorithm uses $N$ copies of a no-external-regret algorithm $\mathcal{A}_1, \ldots, \mathcal{A}_N$, and decides an output distribution computed as a stationary distribution of a Markov chain defined by the output distributions by $\mathcal{A}_1, \ldots, \mathcal{A}_N$. A key idea to improve the genericity and efficiency is the sampling technique to choose an instance from copies that is fed the observed information. In our method, only a chosen instance gets the feedback and updates its output distribution, and the other instances do not update their distributions. This improves computational efficiency and makes the analysis simpler:

meaning better regret bounds are obtained under assumptions on regret bounds with respect to $T$, without input-dependent bounds.

## 2   Related Work

Online decision-making problems with finite actions have been well studied in the context of the expert problem [11], and form the basis for some ensemble learning algorithms including Adaboost [18]. For the problem with full information, the multiplicative weight update (MWU) method [4] achieves $O(\sqrt{T \log N})$-external regret, which is minimax optimal. Indeed, it is known that the external regret is $\Omega(\sqrt{T \log N})$ in the worst case, and hence, there is no room to improve the external regret in terms of worst-case external regret (for details see, e.g., [11]). Some additional assumptions help us to have improved regret bounds. For example, if the cumulative loss $\sum_{t=1}^{T} \ell_i^t$ for each action $i$ is bounded by $M$, $O(\sqrt{M \log N})$-external regret [12; 24], which is called a first-order regret bound, can be achieved. This first-order regret bound is used in [9; 27] to achieve $O(\sqrt{TN \log N})$-swap regret.

Online decision-making with finite action based on bandit information, called multi-armed bandit problems, has been extensively studied not only for stochastic settings but also for adversarial settings [23]. For the adversarial multi-armed bandit problems, Auer et al. [7] showed that MWU approach works well and provides $O(\sqrt{TN \log N})$-external regret. They provided a lower bound of $\Omega(\sqrt{TN})$, which left a gap of an $O(\sqrt{\log N})$-factor. Audibert and Bubeck [5] shaved off this gap by providing an algorithm with $O(\sqrt{TN})$-expected external regret.

Swap regret, which was introduced by [9], is closely related to *internal regret* [15; 16; 17]. Internal regret is defined similarly to swap regret: it is defined as the maximum of $R^T(F)$ over $F \in \Phi_{\text{in}} = \{F_{ij} \mid i, j \in [N], i \neq j\}$ where $F_{ij} : [N] \to [N]$ is defined by $F_{ij}(i) = j$ and $F_{ij}(k) = k$ for $k \in [N] \setminus \{i\}$, while swap regret is defined with $\Phi_{\text{sw}} := \{F : [N] \to [N]\}$, i.e., all functions from $[N]$ to $[N]$. Because it holds that (internal regret) $\leq$ (swap regret) $\leq N \times$ (internal regret), any no-internal-regret algorithms are no-external-regret, and vice versa (for details see, e.g., [11]). Stoltz and Lugosi [35] have introduced a more general notion called $\Phi$-regret, which is defined by $\max_{F \in \Phi} \mathbf{E}[R^T(F)]$ for an arbitrary class $\Phi$ of functions. These have been extensively studied in much literature [3; 19; 22; 26; 27]. Rakhlin et al. [30] provided a more general framework including $\Phi$-regret, which immediately recovers $O(\sqrt{T \log N})$-bounds for internal and external regret and an $O(\sqrt{TN \log N})$-bound for swap regret. Their framework, however, does not provide tight lower bounds for internal and swap regrets.

The importance of swap regret and internal regret is owing to the connection to correlated equilibria in strategic games [8]. Foster and Vohra [16] have shown that, if all players in a strategic game with finite actions follow a no-internal-regret algorithm then their averaged empirical distributions converge to correlated equilibrium. For this connection, a more detailed analysis of computational aspects is given by Hazan and Kale [22]. This connection implies that correlated equilibria are computationally tractable, in contrast to that mixed Nash equilibria are hard to compute [14; 31]. In this context, lower bounds for swap regret provides limitations of the computational efficiency of regret-minimization approaches to correlated equilibrium.

## 3   Problem Statement

Let $N$ and $T$ denote the numbers of actions and time steps. The *repeated decision-making* problem proceeds as follows: In each time step $t \in [T] := \{1, 2, \ldots, T\}$, a player chooses a probability distribution $p^t = (p_1^t, \ldots, p_N^t)^\top \in \Delta^N := \{p \in [0,1]^N : \|p\|_1 = 1\}$ over the action space $[N]$, and picks an action $i^t \in [n]$ following $p^t$. After choosing the action, the player gets feedback regarding the *loss vector* $\ell^t = (\ell_1^t, \ldots, \ell_N^t)^\top \in [0,1]^N$, and incurs the loss of $\ell_{i^t}^t$ for the chosen action. The environment may choose loss vectors $\ell^t$ that depends on the probability distributions $p^1, \ldots, p^t$ and the past actions $i^1, \ldots, i^{t-1}$ chosen by the player.

For *switching functions* $F : [N] \to [N]$, which are arbitrary functions from the action space to itself, we define $R^T(F)$ by $R^T(F) = \sum_{t=1}^{T} (\ell_{i^t}^t - \ell_{F(i^t)}^t)$. The (expected) external regret is defined as the maximum of (the expectation of) $R^T(F)$ over $F \in \Phi_{\text{ex}} = \{F_i \mid [N] \to \{i\} \mid i \in [N]\}$, where $F_i$

denotes the functions that map all actions to single-action $i$, for each $i \in [N]$. The swap regret is defined similarly for $F \in \Phi_{\mathrm{sw}} = \{F : [N] \to [N]\}$, where $\Phi_{\mathrm{sw}}$ is the set of all $N^N$ functions from $[N]$ to $[N]$.

## 4 Proof for Lower Bound

In this section, we provide a proof of Theorem 1. We first describe the environment introduced in [9], in which the player must choose actions in a *balanced* manner in order to avoid large (linear) regret. For this environment, Blum and Mansour [9] showed that the swap regret is $\Omega(\sqrt{TN})$ when the chosen actions are balanced. Under the same condition, we refine their analysis to show $\Omega(\sqrt{TN \log N})$.

### 4.1 Environment construction

We introduce the adaptive environment proposed by Blum and Mansour [9]. We suppose that $T$ and $N$ satisfy the assumptions in Theorem 1. Further, we suppose that $N$ is a multiple of $64$ and that $N$ is a square number for simplicity.[3] Denote $N' = N/2$. For $i \in [N']$ and $t \in [T]$, let $\ell_i'^t \in \{0, 1\}$ follow Bernoulli distributions of parameter $1/2$ independently, i.e., $\ell_i'^t = 0$ with probability $1/2$ and $\ell_i'^t = 1$ with probability $1/2$. For $i \in [N' + 1, 2N']$, fix $\ell_i'^t = 1/2$ for all $t \in [T]$. The actual loss $\ell_i^t$ is identical to $\ell_i'^t$ at the beginning of games, but is set to $1$ if the action $i$ is chosen at least $8T/N$ times so far, i.e., $\ell_i^t$ is defined by

$$\ell_i^t = \begin{cases} \ell_i'^t & \text{if } |\{s \in [t-1] \mid i^s = i\}| < 8T/N, \\ 1 & \text{otherwise} \end{cases}$$

for all $i \in [N]$ and $t \in [T]$. The action whose loss is fixed to $1$ is termed *blocked*.

To analyze the swap regret, we introduce some notations. For each action $i \in [N]$, let $\mathcal{T}_i$ and $n_i$ denote the set of time steps in which the algorithm chooses the $i$-th action and its size, i.e., $\mathcal{T}_i = \{t \in [T] \mid i^t = i\}$ and $n_i = |\mathcal{T}_i|$, respectively. Let $B \subseteq [N]$ denote the set of actions that are blocked at the end of the $T$-th time step, i.e., $B = \{i \in [N] \mid n_i \geq 8T/N\}$. Since $\sum_{i=1}^N n_i = T$, the number $|B|$ of blocked actions is at most $N/8$. For $i, j \in [N]$, let $L_{ij}$ and $L_{ij}'$ denote the sums of $\ell_j^t$ and $\ell_j'^t$ for $t \in \mathcal{T}_i$, respectively, i.e., $L_{ij} = \sum_{t \in \mathcal{T}_i} \ell_j^t$ and $L_{ij}' = \sum_{t \in \mathcal{T}_i} \ell_j'^t$. Since $\ell_j'^t = \ell_j^t$ holds for $j \in [N] \setminus B$ and $t \in [T]$, we have $L_{ij}' = L_{ij}$ if $j \in [N] \setminus B$. By means of these notations, the swap regret can be expressed as follows:

$$\max_F R^T(F) = \max_F \sum_{i=1}^N \sum_{t \in \mathcal{T}_i} (\ell_i^t - \ell_{F(i)}^t) = \sum_{i=1}^N \max_{j \in [N]} \sum_{t \in \mathcal{T}_i} (\ell_i^t - \ell_j^t) = \sum_{t=1}^T \ell_{i^t}^t - \sum_{i=1}^N \min_{j \in [N]} L_{ij}. \quad (1)$$

Since $\mathbf{E}[\ell_i^t] \geq \mathbf{E}[\ell_i'^t] = 1/2$ for all $t \in [T]$ and $i \in [N]$ and since $i^t$ is independent of $\ell'^t$, we have $\mathbf{E}\left[\sum_{t=1}^T \ell_{i^t}^t\right] \geq T/2$. Hence, we have

$$\mathbf{E}\left[\max_F R^T(F)\right] \geq \frac{T}{2} - \mathbf{E}\left[\sum_{i=1}^N \min_{j \in [N]} L_{ij}\right] \geq \frac{T}{2} - \mathbf{E}\left[\sum_{i=1}^N \min_{j \in [N] \setminus B} L_{ij}'\right] \quad (2)$$

where the second inequality follows from the fact that $L_{ij}' = L_{ij}$ for $j \in [N] \setminus B$.

Let $T^u$ denote the number of time steps at which the player chooses a blocked action, i.e.,

$$T^u = \sum_{i=1}^N \max\left\{0, n_i - \frac{8T}{N}\right\} = \sum_{i \in B} \left(n_i - \frac{8T}{N}\right). \quad (3)$$

From the definition of $B$ and $\ell^t$, we have the following result:

**Lemma 1** ([9]). *If* $\mathbf{E}[T^u] \geq T/16$, *we have* $\mathbf{E}[\max_F R^T(F)] \geq T/32$. *If* $\mathbf{E}[T^u] \leq T/16$, *the set* $S \subseteq [N]$ *of actions defined by*

$$S = \{i \in [N] \mid n_i \geq T/(4N)\} \quad (4)$$

*satisfies* $|S| \geq N/16$ *with probability at least* $1/2$.

This Lemma implies, intuitively, that the player needs to avoid choosing blocked actions in order not to suffer large regret, and that, when the player chooses blocked actions at most $T/16$ times, the chosen actions are balanced, i.e., there are $\Omega(N)$ actions that are chosen $\Omega(T/N)$ times.

Blum and Mansour [9] showed $\Omega(\sqrt{TN})$-regret lower bound on the basis of Lemma 1 combined with (2). Their analysis can be summarized as follows: From Lemma 1, it suffices to ponder the case of $\mathbf{E}[T^u] \leq T/16$, and then $|S| \geq N/16$ with a positive probability. For any fixed $\{\mathcal{T}_i\}_{i=1}^N$ such that $|S| \geq N/16$, we can show that $n_i/2 - \mathbf{E}[\min_{j \in [N] \setminus B} L'_{ij}] = \Omega(\sqrt{n_i})$ under some assumption. Indeed, when $\mathcal{T}_i$ is fixed, $L'_{ij} := \sum_{t \in \mathcal{T}_i} \ell'_{ij}$ follows a binomial distribution $Bi(n_i, 1/2)$ independently for $i \in [N]$ and $j \in [N']$. Since $|B| \leq N/8$, the value $\min_{j \in [N'] \setminus B} L'_{ij}$ is at most the $(N/8 + 1)$-th smallest value among $N/2$ independent samples from $Bi(n_i, 1/2)$, which is $n_i/2 - \Omega(\sqrt{n_i})$ with high probability. Combining this discussion and (2), we have $\mathbf{E}\left[\max_F R^T(F)\right] \geq T/2 - \sum_{i=1}^N (n_i/2 - \Omega(\sqrt{n_i})) = \Omega\left(\sum_{i=1}^N \sqrt{n_i}\right)$. Since $n_i = \Omega(T/N)$ for any $i \in S$ from the definition (4) of $S$ and since we may assume $|S| = \Omega(N)$ from Lemma 1, we have $\mathbf{E}\left[\max_F R^T(F)\right] \geq \sum_{i=1}^N \sqrt{n_i} \geq \sum_{i \in S} \sqrt{n_i} = \Omega(N\sqrt{T/N}) = \Omega(\sqrt{TN})$.

## 4.2 Refined analysis

We refine the analysis to present a tighter lower bound. In contrast to the previous work [9] that has evaluated $\min_{j \in [N] \setminus B} L'_{ij}$ separately for each $i$, this subsection analyzes the sum of this for $i \in [N]$. A key lemma for showing the tight lower bound is the following:

**Lemma 2.** *Fix $\{\mathcal{T}_i\}_{i=1}^N$ such that $|S| \geq \frac{N}{16}$. Then, $\sum_{i=1}^N \min_{j \in [N] \setminus B} L'_{ij} \leq \frac{T}{2} - \frac{\sqrt{TN \log N}}{512}$ holds with probability at least $1 - 2\exp(-N^{3/2}/128)$ (w.r.t. the randomness of $(\ell_i'^t)_{i \in [N], t \in T}$).*

We note that Lemma 2 is not about posterior distribution conditioned on $\{\mathcal{T}_i\}_{i=1}^N$, but we regard the value $\sum_{i=1}^N \min_{j \in [N] \setminus B} L'_{ij}$ as a function in $\{\ell_t\}_{t=1}^T$ for a fixed $\{\mathcal{T}_i\}_{i=1}^N$. Hence, we may assume the losses follow i.i.d. Bernoulli distributions in the proof of Lemma 2.

By combining (2), Lemma 1 and Lemma 2, and by using the union bound over all possible choices of $\{\mathcal{T}_i\}_{i=1}^N$, we obtain Theorem 1 as follows.

*Proof of Theorem 1.* From Lemma 1 combined with the assumption of $T \geq 4N \log N$, if $\mathbf{E}[T_u] \geq T/16$, we have $\mathbf{E}[\max_F R^T(F)] \geq T/32 \geq \sqrt{TN \log N}/16$. In the following, we assume $\mathbf{E}[T_u] \leq T/16$. From Lemma 2 and the union bound, since the number of realizable patterns of $\{\mathcal{T}_i\}_i^N$ is at most $N^T$, it holds for arbitrary fixed $\{\mathcal{T}_i\}_{i=1}^N$ satisfying $|S| \geq N/16$ that $\sum_{i=1}^N \min_{j \in [N] \setminus B} L'_{ij} \leq T/2 - \sqrt{TN \log N}/512$, with probability at least $1 - N^T \cdot 2\exp(-N^{3/2}/128) = 1 - 2\exp(-N^{3/2}/128 + T \log N) \geq 3/4$, where the last inequality follows from the assumptions of $T \leq N^{3/2}/(256 \log N)$ and $N \geq 2^{64}$. Since $\{\mathcal{T}_i\}_{i=1}^N$ satisfies $|S| \geq N/16$ with probability at least $1/2$ from Lemma 1, we have $\sum_{i=1}^N \min_{j \in [N] \setminus B} L'_{ij} \leq T/2 - \sqrt{TN \log N}/512$ with probability at least $1/4$. From this and the fact that $\sum_{i=1}^N \min_{j \in [N] \setminus B} L'_{ij} \leq T/2$ holds with probability one, we have $\mathbf{E}\left[\sum_{i=1}^N \min_{j \in [N] \setminus B} L'_{ij} \leq T/2\right] \leq T/2 - \sqrt{TN \log N}/2056$. Combining this and (2), we obtain $\mathbf{E}\left[\max_F R^T(F)\right] \geq \sqrt{TN \log N}/2056$. $\square$

In the remainder of this section, we provide a proof sketch for Lemma 2. Fix $\{\mathcal{T}_i\}_{i=1}^n$ such that $|S| \geq N/16$. Let $\sigma_i : [N'] \to [N']$ be a permutation over $[N']$ such that $L'_{i\sigma_i(1)} \leq L'_{i\sigma_i(2)} \leq \cdots \leq L'_{i\sigma_i(N')}$. If such $\sigma_i$ is not unique, we choose a permutation satisfying the above condition uniformly at random, independently for $i \in [N]$. We denote $\sigma_i[n] = \{\sigma_i(1), \sigma_i(2), \ldots, \sigma_i(n)\} \subseteq [N']$ for $i \in [N]$ and $n \in [N']$. Define $V \subseteq S$ and $U(B') \subseteq [N]$ for $B' \subseteq [N]$ by

$$V = \left\{i \in S \mid L'_{i\sigma_i(\sqrt{N})} \leq \frac{n_i}{2} - \frac{\sqrt{n_i \log N}}{8}\right\}, \quad U(B') = \left\{i \in [N] \mid \sigma_i[\sqrt{N}] \subseteq B'\right\} \quad (5)$$

Using concentration inequalities [10] for independent random variables and anti-concentration inequalities for binomial distributions (see, e.g., Proposition 7.3.2. of [25]), we obtain the following:

**Lemma 3.** *With probability at least $1 - \exp(-N^{3/2}/128)$, We have $|V| \geq 3N/64$.*

**Lemma 4.** *With probability at least* $1 - \exp(-N^{3/2}/128)$*, for any* $B' \subseteq [N']$ *such that* $|B'| \leq N/8$*,* $|U(B')| \leq N/64$ *holds.*

The proofs of these lemmas are provided in the appendix. By combining Lemmas 3 and 4, we obtain Lemma 2 as follows:

*Proof of Lemma 2.* For any $B'$ and $i$, if $\sigma_i[\sqrt{N}]$ is not included in $B'$, we have $\min_{j \in [N] \setminus B'} L_{ij} \leq L_{i\sigma_i(\sqrt{N})}$. Hence, from the definition of $V$ and $U(B')$, for any $i \in V \setminus U(B')$, we have $\min_{j \in [N] \setminus B'} L'_{ij} \leq n_i/2 - \sqrt{n_i \log N}/8$. Combining this and the fact that $\min_{j \in [N] \setminus B} L'_{ij} \leq n_i/2$ for all $i \in [N]$, we obtain

$$
\begin{aligned}
\sum_{i=1}^{N} \min_{j \in [N] \setminus B} L'_{ij} &= \sum_{i \in V \setminus U(B)} \min_{j \in [N] \setminus B} L'_{ij} + \sum_{i \in [N] \setminus (V \setminus U(B))} \min_{j \in [N] \setminus B} L'_{ij} \\
&\leq \sum_{i \in V \setminus U(B)} \left( \frac{n_i}{2} - \frac{\sqrt{n_i \log N}}{8} \right) + \sum_{i \in [N] \setminus (V \setminus U(B))} \frac{n_i}{2} = \frac{T}{2} - \sum_{i \in V \setminus U(B)} \frac{\sqrt{n_i \log N}}{8} \\
&\leq \frac{T}{2} - |V \setminus U(B)| \cdot \frac{1}{8} \sqrt{\frac{T \log N}{4N}} = \frac{T}{2} - \frac{|V \setminus U(B)|}{16} \sqrt{\frac{T \log N}{N}}
\end{aligned}
$$

where the second equality follows from $\sum_{i=1}^{N} n_i = 1$, and the last inequality follows from $V \subseteq S$ and the definition (4) of $S$. From Lemmas 3 and 4, it holds with probability $1 - 2\exp(-N^{3/2}/128)$ that $|V \setminus U(B)| \leq N/32$. Combining this and the above, we obtain Lemma 2. □

# 5 Randomized Efficient Reduction from External to Swap Regret

This section provides a generic and efficient reduction method that offers a no-swap-regret algorithm given no-external-regret algorithms. Such methods have been provided by Blum and Mansour [9], which achieve swap regret of $O(N \cdot r(T))$,[4] given an $r(T)$-external-regret algorithm, for the full-information setting as well as for the bandit setting. Our reduction method provides an $O(N \cdot r(T/N))$-swap-regret bound. If $r(T) = \Theta(\sqrt{T})$ (ignoring dependency on $N$), our swap-regret bound is $O(\sqrt{N} \cdot r(T))$. which matches the tight regret bound in the full-information setting, and improves over the state-of-the-art in the bandit setting.

Our reduction method is similar to one by Blum and Mansour [9], except for that ours employs a *two-step randomization* technique. The method starts with instantiating $N$ copies $\mathcal{A}_1, \ldots, \mathcal{A}_N$ of no-external-regret algorithms $\mathcal{A}$. For all $i \in [N]$, let $t_i$ denote the current time step for $\mathcal{A}_i$, and let $q_i^{t_i} \in \Delta^N$ be its output distribution. Let $p^t = (p_1^t, \ldots, p_N^t)^\top \in \Delta^N$ to be a distribution satisfying

$$
p^t = \sum_{i=1}^{N} p_i^t q_i^{t_i}. \tag{6}
$$

Such $p^t$ can be interpreted as a stationary distribution of a Markov chain over $[N]$ of which transition probabilities are given by $(q_1^{t_1}, \ldots, q_N^{t_N})$, and can be computed efficiently [13]. In the reduction method by Blum and Mansour [9], after returning $i^t \sim p^t$ and getting the feedback of $\ell^t$, each instance $\mathcal{A}_i$ is fed the loss of $p_i^t \ell^t$ and the time step $t_i$ is incremented for all $i \in [N]$. Our method adopts a different strategy to feed the loss to no-external-regret algorithms: We pick $j^t \in [N]$ from the distribution $p^t$, and then, after returning the action $i^t \sim q_j^{t_j}$ for $j = j^t$, feed the loss $\ell^t$ (or $\ell_{i^t}^t$ in the bandit setting) to $\mathcal{A}_{j^t}$. Instances $\mathcal{A}_i$ for $i \in [N] \setminus \{j^t\}$ are not updated in this time step. We note that, from (6), $i^t$ follows $p^t$ marginalizing $j^t \sim p^t$. Our method is summarized in Algorithm 1. The output of Algorithm 1 has the following regret bound:

**Lemma 5.** *Suppose that* $\mathcal{A}$ *is an* $r(T)$-*external-regret algorithm, i.e., for arbitrary environments, if* $k^t$ *is chosen by* $\mathcal{A}$*, it hold for all* $i^* \in [N]$ *and* $T$ *that* $\mathbf{E}\left[ \sum_{t=1}^{T} (\ell_{k^t}^t - \ell_{i^*}^t) \right] \leq r(T)$*. Then, for*

**Algorithm 1** Reduction from external to swap regret
---
**Require:** No-external-regret algorithm $\mathcal{A}$, the number $N$ of actions.
1: Instantiate $N$ copies $\mathcal{A}_1, \ldots, \mathcal{A}_N$ of $\mathcal{A}$, and initialize their time step by $t_1 = \cdots = t_N = 1$.
2: **for** $t = 1, 2, \ldots$ **do**
3:     Compute $p^t$ satisfying (6)
4:     Choose $j^t \sim p^t$.
5:     Return $i^t \sim q_j^{t_j}$ where $j = j^t$, and get the feedback.
6:     Feed the observed loss ($\ell^t$ in the full-information setting, or $\ell_{i^t}^t$ in the bandit setting) into $\mathcal{A}_{j^t}$, increment $t_{j^t}$, and update the output distribution of $\mathcal{A}_{j^t}$.
7: **end for**
---

*arbitrary $F : [N] \to [N]$ and $T$, the output of Algorithm 1 satisfies*

$$\mathbf{E}\left[\sum_{t=1}^{T}(\ell_{i^t}^t - \ell_{F(i^t)}^t)\right] \leq \max\left\{\sum_{i=1}^{N} r(T_i) \;\middle|\; T_i \in \mathbb{Z}_{\geq 0}, \sum_{i=1}^{N} T_i = T\right\}. \tag{7}$$

*Proof.* Since $i^t$ and $j^t$ follows the same distribution $p^t$ from (6), it holds for any fixed $F$ that

$$\mathbf{E}\left[\sum_{t=1}^{T}(\ell_{i^t}^t - \ell_{F(i^t)}^t)\right] = \mathbf{E}\left[\sum_{t=1}^{T}(\ell_{i^t}^t - \ell_{F(j^t)}^t)\right] = \mathbf{E}\left[\sum_{i=1}^{N}\sum_{t \in \mathcal{U}_i}(\ell_{i^t}^t - \ell_{F(i)}^t)\right], \tag{8}$$

where we define $\mathcal{U}_i = \{t \in [T] \mid j^t = i\}$. Fix $i \in [N]$ arbitrarily. In all time steps $t \in \mathcal{U}_i$, the conditional distribution of $i^t$ given $j^t$ is determined by $\mathcal{A}_i$, and the loss is fed into $\mathcal{A}_i$. Hence, from the assumption of Lemma 5, we have $\mathbf{E}\left[\sum_{t \in \mathcal{U}_i}(\ell_{i^t}^t - \ell_{F(i)}^t)\right] \leq \mathbf{E}\left[r(|\mathcal{U}_i|)\right]$ for arbitrary $F$. Combining this and (8), we have $\mathbf{E}\left[\sum_{t=1}^{T}(\ell_{i^t}^t - \ell_{F(i^t)}^t)\right] \leq \mathbf{E}\left[\sum_{i=1}^{N} r(|\mathcal{U}_i|)\right]$. Since $\sum_{i=1}^{N} |\mathcal{U}_i| = T$ from the definition of $\mathcal{U}_i$, we get the bound of (7). $\square$

This lemma immediately leads to the regret bound in Theorem 2.

*Proof of Theorem 2.* Assuming $r(T)$ be a concave function, we can bound the right-hand side of (7) as follows: $\sum_{i=1}^{N} r(T_i) = N \cdot \frac{1}{N}\sum_{i=1}^{N} r(T_i) \leq N \cdot r\left(\frac{1}{N}\sum_{i=1}^{N} T_i\right) = N \cdot r\left(\frac{T}{N}\right)$, where we applied Jensen's inequality with the aid of the assumption that $r$ is concave.

The computational time of Algorithm 1 depends on that of $\mathcal{A}$. Let $\mathsf{Init}_{\mathcal{A}}$ and $\mathsf{Step}_{\mathcal{A}}$ to denote the times taken to initialize $\mathcal{A}$, and to update the output distribution of $\mathcal{A}$, respectively. In the first step of Algorithm 1, the computational time of $N \cdot \mathsf{Init}_{\mathcal{A}}$ is consumed to initialize $\mathcal{A}_1, \ldots, \mathcal{A}_N$. In each iteration, we compute a stationary distribution of a Markov chain with $N$ states, which consumes $\mathsf{SD}_N$ time, and update the output distribution of one of instances $\mathcal{A}_1, \ldots, \mathcal{A}_N$, which requires $\mathsf{Step}_{\mathcal{A}}$ time. Summarizing the above, the overall time complexity of Algorithm 1 is bounded by $N \cdot \mathsf{Init}_{\mathcal{A}} + T \cdot (\mathsf{Step}_{\mathcal{A}} + \mathsf{SD}_N)$. $\square$

Theorem 2 implies that there exists a computationally efficient algorithm that achieves swap regret of $O(\sqrt{TN^2})$ in the bandit setting. In fact, when we choose $\mathcal{A}$ to be computationally efficient multi-armed bandit algorithms with expected external regret bounded by $O(\sqrt{TN})$, e.g., the ones by Audibert and Bubeck [5]; Zimmert and Seldin [36], Algorithm 1 leads to the following:

**Corollary 1.** *In the bandit setting, there is a polynomial-time algorithm that achieves* $\max_F \mathbf{E}\left[\sum_{t=1}^{T}(\ell_{i^t}^t - \ell_{F(i^t)}^t)\right] = O(\sqrt{TN^2})$.

## 6 Conclusion

In this paper, we showed an $\Omega(\sqrt{TN\log N})$-lower bound for the swap regret, which is tight up to constant in the full-information setting. We also provided a computationally efficient algorithm for swap-regret minimization, which can be applied to the full-information setting as well as the bandit setting. As can be seen in Table 1, the tight bound for swap regret in the bandit setting is left as an

open question. To solve this, the technique for external-regret lower bounds [6] may be helpful. As mentioned in Stoltz [33], tight bounds for internal regret would be an interesting question as well.

## Broader impact

The authors believe that this paper presents neither ethical nor societal issues, as this is a theoretical work.

## Acknowledgments and Disclosure of Funding

The author thanks Kaito Fujii and Ayumi Igarashi for valuable discussions. The author also thanks anonymous reviewers, especially a reviewer for ICML2020 who pointed out a technical error in the previous manuscript, for many helpful comments and suggestions, The author was supported by JST, ACT-I, Grant Number JPMJPR18U5, Japan.

## Footnotes

[1] $\mathsf{Init}_\mathcal{A}$ and $\mathsf{Step}_\mathcal{A}$ mean $\mathcal{A}$'s time complexity of the initialization and that per time step, respectively.

[2] We may assume $\mathsf{SD}_N$ is almost $O(N^2)$. In fact, as shown by Cohen et al. [13], we can approximately compute a stationary distribution of a given Markov chain in $\tilde{O}(m + N^{1+o(1)})$-time ignoring logarithmic factors, where $m$ represents the number of nonzero entries of its transition matrix.

[3]This assumption can easily be relaxed.

[4]This can be improved to $O(\sqrt{N} \cdot r(T))$ under the assumption of the first-order regret bound, only for the full-information setting, as shown in [9; 27].

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
