[Supplementary Material]

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

# A Proof of lemmas

## A.1 Proof of Lemma 1

*Proof.* Since $\ell_i'^t$ is independent of the player's action $i^t$, and since $\mathbf{E}[\ell_i'^t] = 1/2$ for all $i \in [N]$, we have $\mathbf{E}[\ell_{i^t}'^t] = 1/2$. Hence, if $i^t$ is not blocked at the time step $t$, we have $\mathbf{E}[\ell_{i^t}^t] = 1/2$. Further, if $i^t$ is blocked at the time step $t$, we have $\ell_{i^t}^t = 1$ from the definition of $\ell_{i^t}^t$. Hence, the cumulative loss for the player can be expressed as

$$\mathbf{E}\left[\sum_{t=1}^{T} \ell_{i^t}^t\right] = \mathbf{E}\left[(T - T^u) \cdot \frac{1}{2} + T^u \cdot 1\right] = \frac{T}{2} + \frac{1}{2}\mathbf{E}[T^u]. \tag{9}$$

On the other hand, we have

$$\mathbf{E}\left[\sum_{i=1}^{N} \min_{j \in [N]} L_{ij}\right] \leq \mathbf{E}\left[\sum_{i=1}^{N} \min_{j \in [N]\setminus B} L_{ij}'\right] \leq \mathbf{E}\left[\sum_{i=1}^{N} \frac{n_i}{2}\right] = \frac{T}{2}, \tag{10}$$

where the first inequality follows from $L_{ij} = L_{ij}'$ for $j \in [N] \setminus B$, and the second inequality comes from the fact that $|B| \leq N/8$ and that $L_{ij}' = n_i/2$ for $j \in [N] \setminus [N']$, which follows from the definition of $\ell_i'^t$. Combining (1), (9), and (10), we obtain

$$\mathbf{E}\left[\max_F R^T(F)\right] = \mathbf{E}\left[\sum_{t=1}^{T} \ell_{i^t}^t - \sum_{i=1}^{N} \min_{j \in [N]} L_{ij}\right] \geq \mathbf{E}\left[\frac{T}{2} + \frac{T^u}{2} - \frac{T}{2}\right] \geq \frac{\mathbf{E}[T^u]}{2}.$$

From this, if $\mathbf{E}[T^u] \geq T/16$, we have $\mathbf{E}\left[\max_F R^T(F)\right] \geq T/32$. Next, we assume that $\mathbf{E}[T^u] \leq T/16$. Then, from Markov's inequality, $T^u \leq T/8$ holds with probability at least $1/2$. When $T^u \leq T/8$ holds, $S \subseteq [N]$ defined in (4) satisfies $|S| \geq N/16$. Indeed, we have

$$T^u \geq \sum_{i \in S} \max\{0, n_i - 8T/N\} \geq \sum_{i \in S} n_i - \frac{8T|S|}{N} = T - \sum_{i \in [N]\setminus S} n_i - \frac{8T(|S|)}{N}$$

$$\geq T - \frac{T(N - |S|)}{4N} - \frac{8T(|S|)}{N} = T\left(\frac{3}{4} - \frac{31|S|}{4N}\right),$$

where the first inequality comes from (3), the first equality follows from $\sum_{i=1}^{N} n_i = T$, and the last inequality holds since $n_i < T/(4N)$ for $i \in [N] \setminus S$ follows from the definition (4) of $S$. From the above inequality and the assumption of $T^u \leq T/8$, we have $\frac{31|S|}{4N} \geq \frac{1}{2}$, which implies $|S| > N/16$. □

## A.2 Preliminary for Lemmas 3 and 4

In this subsection, we provides (anti-)concentration inequalities that are used in the proof of Lemmas 3 and 4.

**Lemma 6** (Section 2.13 of [10],[2; 29]). *Let $X$ be a random variable following a binomial distribution $Bi(n, p)$, where $p \in (0, 1)$. Then, for any $\lambda \geq 0$, we have*

$$\text{Prob}[X \leq np - \lambda] \leq \exp\left(-\frac{\lambda^2}{2np}\right).$$

*Proof.* The random variable $X$ can be expressed as $X = \sum_{i=1}^{n} Y_i$, where $Y_i$ follows a Bernoulli distribution of parameter $p$, respectively. Then, for arbitrary $s > 0$, we have

$$\text{Prob}[X \leq np - \lambda] = \text{Prob}\left[\sum_{i=1}^{n}(Y_i - p) \leq -\lambda\right] = \text{Prob}\left[\exp\left(-s\sum_{i=1}^{n}(Y_i - p)\right) \geq \exp(s\lambda)\right]$$

$$\leq \frac{1}{\exp(s\lambda)} \mathbf{E}\left[\exp\left(-s\sum_{i=1}^{n}(Y_i - p)\right)\right] = \frac{1}{\exp(s\lambda)}\left(\mathbf{E}\left[\exp(-s(Y_1 - p))\right]\right)^n,$$

where the inequality follows from Markov's inequality, and the last equality holds since $Y_1, \ldots, Y_n$ follow i.i.d. distributions. We can bound $\mathbf{E}[\exp(-s(Y_1 - p))]$ as follows:

$$\mathbf{E}\left[\exp(-s(Y_1 - p))\right] = p\exp(-s(1-p)) + (1-p)\exp(ps) = \exp(ps)(p\exp(-s) + 1 - p)$$

$$\leq \exp(ps)\left(p\left(1 - s + \frac{s^2}{2}\right) + 1 - p\right) = \exp(ps)\left(1 - ps + \frac{ps^2}{2}\right)$$

$$\leq \exp(sp)\exp\left(-ps + \frac{ps^2}{2}\right) = \exp\left(\frac{ps^2}{2}\right),$$

where the first inequality follows from $\exp(-x) \leq 1 - x + x^2/2$ for $x \leq 0$, and the second inequality comes from $1 + x \leq \exp(x)$ for $x \in \mathbb{R}$. Combining the above two displayed inequalities, we obtain

$$\mathrm{Prob}\left[X \leq np - \lambda\right] \leq \exp\left(\frac{nps^2}{2} - s\lambda\right) = \exp\left(\frac{np}{2}\left(s - \frac{\lambda}{np}\right)^2 - \frac{\lambda^2}{2np}\right).$$

Since this inequality holds for arbitrary $s > 0$, by setting $s = \frac{\lambda}{2np}$ we obtain the first inequality in Lemma 6. $\qquad\square$

**Lemma 7.** *Let $X_1, X_2, \ldots, X_n$ be independent random variables following Bernoulli distributions of parameters $p_1, p_2, \ldots, p_n$ respectively, i.e., $X_i = 1$ with probability $p_i$ and $X_i = 0$ with probability $(1 - p_i)$ for each $i \in [n]$. Suppose that there exists $p \in [0, 1]$ for which $p_i$ are bounded as $p_i \leq p$ for all $i \in [n]$. Then, for any $k \in [n]$, we have*

$$\mathrm{Prob}\left[\sum_{i=1}^{n} X_i \geq k\right] \leq 2^n p^k.$$

*Proof.* By using the union-bound technique, we obtain the following:

$$\mathrm{Prob}\left[\sum_{i=1}^{n} X_i \geq k\right] = \mathrm{Prob}\left[\exists S \subseteq [n], \quad |S| \geq k, \quad \forall i \in S, \quad X_i = 1\right]$$

$$\leq \sum_{S \subseteq [n]: \, |S| \geq k} \mathrm{Prob}\left[\forall i \in S, X_i = 1\right] = \sum_{S \subseteq [n]: \, |S| \geq k} \prod_{i \in S} p_i$$

$$\leq \sum_{S \subseteq [n]: \, |S| \geq k} p^{|S|} \leq \sum_{S \subseteq [n]: \, |S| \geq k} p^k \leq 2^n p^k,$$

where the first inequality follows from the union bound, the second equality follows from the assumption that $X_1, X_2, \ldots, X_n$ are independent, and the second inequality follows from the assumption of $p_i \leq p$. $\qquad\square$

**Lemma 8.** *Let $k$ and $m$ be positive integers satisfying $2k \leq m + 1$. We then have*

$$\binom{2m}{m-k} \geq \frac{2^{2m}}{2\sqrt{m}}\exp\left(-\frac{2\log 2 \cdot k^2}{m+1}\right), \quad \binom{2m-1}{m-k} \geq \frac{2^{2m-1}}{2\sqrt{m}}\exp\left(-\frac{2\log 2 \cdot k(k-1)}{m+1}\right).$$

$$(11)$$

*Proof.* From the definition of binomial coefficients, we have

$$\binom{2m}{m-k} = \frac{(2m)!}{(m-k)!(m+k)!} = \frac{(2m)!}{m!m!}\frac{m!}{(m-k)!}\frac{m!}{(m+k)!} = \binom{2m}{m}\prod_{i=1}^{k}\frac{m-k+i}{m+i}$$

$$= \binom{2m}{m}\prod_{i=1}^{k}\left(1 - \frac{k}{m+i}\right) \geq \binom{2m}{m}\prod_{i=1}^{k}\left(1 - \frac{k}{m+1}\right) = \binom{2m}{m}\left(1 - \frac{k}{m+1}\right)^k$$

$$\geq \binom{2m}{m}\exp\left(-\frac{2\log 2 \cdot k}{m+1}\right)^k = \binom{2m}{m}\exp\left(-\frac{2\log 2 \cdot k^2}{m+1}\right),$$

where the second inequality follows from $1 - x \geq \exp(-2\log 2 \cdot x)$ for $x \in [0, 1/2]$ and that $0 \leq k/(m+1) \leq 1/2$. The binary coefficient $\binom{2m}{m}$ is bounded from below by $\frac{2^{2m}}{2\sqrt{m}}$. Indeed, we have

$$\binom{2m}{m} = \frac{(2m)!}{m!m!} = \prod_{i=1}^{m} \frac{2i-1}{i} \cdot \prod_{i=1}^{m} \frac{2i}{i} = 2^{2m} \prod_{i=1}^{m} \left(\frac{2i-1}{2i}\right) \geq 2^{2m} \cdot \frac{1}{2} \prod_{i=2}^{m} \sqrt{\frac{i-1}{i}} = \frac{2^{2m}}{2\sqrt{m}},$$
(12)

where the inequality can be confirmed as $\left(\frac{2i-1}{2i}\right)^2 = \frac{4i^2 - 4i + 1}{4i^2} \geq \frac{4i^2 - 4i}{4i^2} = \frac{i-1}{i}$. Combining the above two inequalities, we obtain the first inequality in (11). Similarly, we have

$$\begin{aligned}
\binom{2m-1}{m-k} &= \binom{2m-1}{m} \frac{(m-1)!}{(m-k)!} \frac{m!}{(m+k-1)!} \\
&= \binom{2m-1}{m} \prod_{i=1}^{k-1} \frac{m-k+i}{m+i} = \binom{2m-1}{m} \prod_{i=1}^{k-1} \left(1 - \frac{k}{m+1}\right) \\
&\geq \binom{2m-1}{m} \exp\left(-\frac{2\log 2 \cdot k(k-1)}{m+1}\right) \geq \frac{2^{2m-1}}{2\sqrt{m}} \exp\left(-\frac{2\log 2 \cdot k(k-1)}{m+1}\right),
\end{aligned}$$

where the last inequality follows from

$$\binom{2m-1}{m} = \prod_{i=1}^{m} \frac{2i-1}{i} \cdot \prod_{i=1}^{m-1} \frac{2i}{i} = 2^{2m-1} \prod_{i=1}^{m} \left(\frac{2i-1}{2i}\right) \geq 2^{2m-1} \cdot \frac{1}{2} \prod_{i=2}^{m} \sqrt{\frac{i-1}{i}} = \frac{2^{2m-1}}{2\sqrt{m}}.$$

$\square$

**Lemma 9** (Propostion 7.3.2. of [25])**.** *Let $X$ be a random variable following a binomial distribution $Bi(n, 1/2)$. For any $\lambda \in [0, n/8]$, we have*

$$\mathrm{Prob}\left[X \leq \frac{n}{2} - \lambda\right] \geq \frac{1}{15} \exp\left(-\frac{16\lambda^2}{n}\right).$$

*Proof.* We first suppose that $n$ is even, and denote $n = 2m$. Let $r$ be an arbitrary integer such that $2r \leq m+1$. If $X \sim B(n, 1/2)$, we have

$$\begin{aligned}
\mathrm{Prob}\left[X \leq \frac{n}{2} - \lambda\right] &= \frac{1}{2^n} \sum_{k=\lceil\lambda\rceil}^{m} \binom{2m}{m+k} \geq \frac{1}{2^n} \sum_{k=\lceil\lambda\rceil}^{r} \binom{2m}{m+k} \\
&\geq \frac{1}{2\sqrt{m}} \sum_{k=\lceil\lambda\rceil}^{r} \exp\left(-\frac{2\log 2 \cdot k^2}{m+1}\right) \geq \frac{r - \lceil\lambda\rceil + 1}{2\sqrt{m}} \exp\left(-\frac{2\log 2 \cdot r^2}{m+1}\right),
\end{aligned}$$
(13)

where the second inequality follows from (11). Let $r = \lceil\lambda\rceil - 1 + \lceil\sqrt{m}/4\rceil$. We then have $2r \leq m+1$ from the assumption of $\lambda \in [0, n/8]$ and $n = 2m$. Hence, from (13), we have

$$\mathrm{Prob}\left[X \leq \frac{n}{2} - \lambda\right] \geq \frac{1}{8} \exp\left(-\frac{2\log 2 \cdot r^2}{m+1}\right) \geq \frac{1}{8} \exp\left(-\frac{4\log 2 \cdot (\lambda+1)^2}{m} - \frac{\log 2}{4}\right), \quad (14)$$

where the last inequality follows from

$$r^2 \leq \left(\lambda + 1 + \frac{\sqrt{m}}{4}\right)^2 \leq 2\left((\lambda+1)^2 + \left(\frac{\sqrt{m}}{4}\right)^2\right) = 2(\lambda+1)^2 + \frac{m}{8}.$$

Since we have $4\log 2 \cdot (\lambda+1)^2/m + \log 2/4 \leq 8\lambda^2/m + 1/2$ under the condition of $\lambda \geq 2$ or $m \geq 32$, (14) implies

$$\begin{aligned}
\mathrm{Prob}\left[X \leq \frac{n}{2} - \lambda\right] &\geq \frac{1}{8} \exp\left(-\frac{4\log 2 \cdot (\lambda+1)^2}{m} - \frac{\log 2}{4}\right) \\
&\geq \frac{1}{8} \exp\left(-\frac{8\lambda^2}{m} - \frac{1}{2}\right) \geq \frac{1}{15} \exp\left(-\frac{8\lambda^2}{m}\right)
\end{aligned}$$
(15)

if $\lambda \leq 2$ or $m \geq 32$ hold. Otherwise, i.e., if we assume $\lambda < 2$ and $m < 32$, we have

$$\text{Prob}\left[X \leq \frac{n}{2} - \lambda\right] \geq \frac{1}{2\sqrt{m}} \exp\left(-\frac{2\log 2 \cdot (\lceil\lambda\rceil)^2}{m+1}\right) \geq \frac{1}{15}, \tag{16}$$

where the first inequality follows from (13) with $r = \lceil\lambda\rceil$ and the second inequality comes from the assumption of $0 \leq \lambda \leq n/8 = m/4$ and can be confirmed by a simple calculation. From (15) and (16), we have

$$\text{Prob}\left[X \leq \frac{n}{2} - \lambda\right] \geq \frac{1}{15} \exp\left(-\frac{8\lambda^2}{m}\right) = \frac{1}{15} \exp\left(-\frac{16\lambda^2}{n}\right)$$

for all $\lambda \in [0, n/8]$, assuming $n$ is even. Next, we consider the case of odd $n$. Denote $n = 2m - 1$. Then we have

$$\begin{aligned}
\text{Prob}\left[X \leq \frac{n}{2} - \lambda\right] &= \frac{1}{2^n} \sum_{k=\lceil\lambda+1/2\rceil}^{m} \binom{2m-1}{m-k} \geq \frac{1}{2^n} \sum_{k=\lceil\lambda+1/2\rceil}^{r} \binom{2m-1}{m-k} \\
&\geq \frac{1}{2\sqrt{m}} \sum_{k=\lceil\lambda+1/2\rceil}^{r} \exp\left(-\frac{2\log 2 \cdot k(k-1)}{m+1}\right) \\
&\geq \frac{r - \lceil\lambda+1/2\rceil + 1}{2\sqrt{m}} \exp\left(-\frac{2\log 2 \cdot r(r-1)}{m+1}\right) \tag{17}
\end{aligned}$$

for arbitrary integer $r$ such that $2r \leq m + 1$. Let $r = \lceil\lambda + 1/2\rceil - 1 + \lceil\sqrt{m}/4\rceil$. We then have $2r \leq m + 1$ from the assumption of $\lambda \in [0, n/8]$ and $n = 2m - 1$. Hence, from (17), we have

$$\text{Prob}\left[X \leq \frac{n}{2} - \lambda\right] \geq \frac{1}{8} \exp\left(-\frac{2\log 2 \cdot r(r-1)}{m+1}\right) \geq \frac{1}{8} \exp\left(-\frac{4\log 2 \cdot (\lambda+1)^2}{m} - \frac{\log 2}{4}\right), \tag{18}$$

where the second inequality follows from

$$r(r-1) \leq \left(r - \frac{1}{2}\right)^2 \leq \left(\lambda + 1 + \frac{\sqrt{m}}{4}\right)^2 \leq 2(\lambda+1)^2 + \frac{m}{8}.$$

Assuming $\lambda \geq 2$ or $m \geq 32$,

$$\begin{aligned}
\text{Prob}\left[X \leq \frac{n}{2} - \lambda\right] &\geq \frac{1}{8} \exp\left(-\frac{4\log 2 \cdot (\lambda+1)^2}{m} - \frac{\log 2}{4}\right) \\
&\geq \frac{1}{8} \exp\left(-\frac{8\lambda^2}{m} - \frac{1}{2}\right) \geq \frac{1}{15} \exp\left(-\frac{16\lambda^2}{n}\right) \tag{19}
\end{aligned}$$

where the first inequality follows from (18) and the second inequality follows from the assumption of $\lambda \geq 2$ or $m \geq 32$, and the last inequality follows from $n = 2m - 1$ and $\exp(-1/2) \geq 8/15$. If $\lambda < 2$ and $m < 32$, we have

$$\text{Prob}\left[X \leq \frac{n}{2} - \lambda\right] \geq \frac{1}{2\sqrt{m}} \exp\left(-\frac{2\log 2 \cdot \lceil\lambda+1/2\rceil\lceil\lambda-1/2\rceil}{m+1}\right) \geq \frac{1}{15} \tag{20}$$

where the first inequality follows from (17) with $r = \lceil r + 1/2\rceil$ and the last inequality follows from the assumption of $\lambda \leq n/8 = (2m-1)/8$ and a simple calculation. From (19) and (20), we obtain the desired inequality for the case of odd $n$. □

**Lemma 10.** *Let $X_1, \ldots, X_N$ be independent random variables following binomial distributions $Bi(n, 1/2)$. Suppose that $k \in [N]$ and $\alpha$ satisfy $1 < \alpha < \exp(n/4)$ and $15\alpha(k-1) \leq N$. Let $X_{\sigma(k)}$ be the $k$-th smallest value among $X_1, \ldots, X_N$. Then, with probability at least $1 - \exp(k - 1 - N/(30\alpha))$, it holds that $X_{\sigma(k)} < n/2 - \sqrt{n\log\alpha}/4$.*

*Proof.* From Lemma 9, we have

$$p := \text{Prob}\left[X_i \leq \frac{n}{2} - \frac{\sqrt{n\log\alpha}}{4}\right] \geq \frac{1}{15\alpha} \tag{21}$$

for all $i \in [N]$. Define $S \subseteq [N]$ by $S = \{i \in [N] \mid X_i \leq n/2 - \sqrt{n \log \alpha}/4\}$. Then $|S|$ follows a binomial distribution $Bi(n, p)$ since $X_i$ are independent for all $i \in [N]$. Since $X_{\sigma(k)} > n/2 - \sqrt{n \log \alpha}/4$ if and only if $|S| < k$, we obtain

$$\mathrm{Prob}\left[X_{\sigma(k)} > \frac{n}{2} - \frac{\sqrt{n \log \alpha}}{4}\right] = \mathrm{Prob}\left[|S| < k\right] \leq \exp\left(-\frac{(k-1-Np)^2}{2Np}\right)$$

$$\leq \exp\left(k - 1 - \frac{Np}{2}\right) \leq \exp\left(k - 1 - \frac{N}{30\alpha}\right),$$

where the first inequality follows from Lemma 6, and the last inequality follows from (21). □

### A.3 Proof of Lemma 3

*Proof.* Since $L_{ij}$ follow binomial distributions $Bi(n_i, 1/2)$ independently for $i \in [N]$ and $j \in [N']$, from Lemma 10 with $\alpha = N^{1/4}$ and $k = \sqrt{N}$, we have

$$\mathrm{Prob}\left[L'_{i\sigma_i(\sqrt{N})} \geq \frac{n_i}{2} - \frac{\sqrt{n_i \log N}}{8}\right] \leq \exp\left(\sqrt{N} - \frac{N'}{30N^{1/4}}\right) \leq \exp\left(-\frac{N^{3/4}}{120}\right), \quad (22)$$

where the last inequality follows from $N' = N/2$ and the assumption of $N \geq 2^{64}$. Since $\{L'_{i\sigma_i(\sqrt{N})}\}_{i \in S}$ are independent, from (22) and Lemma 7, we have

$$\mathrm{Prob}\left[|V| < \frac{3N}{64}\right] = \mathrm{Prob}\left[|S \setminus V| \geq |S| - \frac{3N}{64}\right]$$

$$\leq \mathrm{Prob}\left[\left|\left\{i \in S \mid L'_{i\sigma(\sqrt{N})} \geq \frac{n_i}{2} - \frac{\sqrt{n_i \log N}}{8}\right\}\right| \geq \frac{N}{64}\right]$$

$$\leq 2^{|S|} \exp\left(-\frac{N}{64} \cdot \frac{N^{3/4}}{120}\right) \leq \exp\left(|S| - \frac{N}{64} \cdot \frac{N^{3/4}}{120}\right) \leq \exp\left(-\frac{N^{3/2}}{128}\right),$$

where the first inequality follows from $|S| \geq N/16$ and the definition (5) of $V$, the second inequality follows from (22) and Lemma 7, and the last inequality follows from $|S| \leq N$ and the assumption of $N \geq 2^{64}$. □

### A.4 Proof of Lemma 4

*Proof.* Since $\sigma_i$ follows a uniform distribution over the set of all permutations over $[N']$, $\sigma_i[\sqrt{N}]$ follows a uniform distribution over $\binom{[N']}{\sqrt{N}}$. Hence, for any fixed $B'$ such that $|B'| \leq N/8$, we have

$$\mathrm{Prob}\left[\sigma_i[\sqrt{N}] \subseteq B'\right] \leq \frac{\binom{|B'|}{\sqrt{N}}}{\binom{N'}{\sqrt{N}}} \leq \left(\frac{|B'|}{N'}\right)^{\sqrt{N}} \leq \left(\frac{1}{4}\right)^{\sqrt{N}} \leq \exp(-\sqrt{N}), \quad (23)$$

where the third inequality follows from $|B'| \leq N/8$ and $N' = N/2$. Since $\{\sigma_i[\sqrt{N}]\}_{i \in [N]}$ are independent, from Lemma 7, it holds for fixed $B' \subseteq [N]$ with $|B'| \leq N/8$ that

$$\mathrm{Prob}\left[|U(B')| \geq \frac{N}{64}\right] = \mathrm{Prob}\left[|\{i \in [N] \mid \sigma_i[\sqrt{N}] \subseteq B'\}| \geq \frac{N}{64}\right]$$

$$\leq 2^N \exp\left(-\sqrt{N} \cdot \frac{N}{64}\right) \leq \exp\left(N - \frac{N^{3/2}}{64}\right).$$

Since this holds for any $B' \subseteq [N]$ such that $|B'| \leq N/8$, by the union bound, we have

$$\mathrm{Prob}\left[\exists B' \subseteq [N], \quad |B'| \leq \frac{N}{8}, \quad |U(B')| \geq \frac{N}{64}\right] \leq 2^N \exp\left(N - \frac{N^{3/2}}{64}\right)$$

$$\leq \exp\left(2N - \frac{N^{3/2}}{64}\right) \leq \exp\left(-\frac{N^{3/2}}{128}\right),$$

where the last inequality follows from the assumption of $N \geq 2^{64}$. □