[Reviews · NeurIPS 2020]

Review 1

Summary and Contributions: The paper studies swap regret for the problem of prediction with expert advice. This is the difference between total loss of the learner and that of any comparator that can replace any action played by the learner with any other fixed action. This work shows a lower bound on the swap regret of \sqrt{T N log N}, where T is the time horizon and N is the number of actions, matching the upper bound of Blum and Mansour. Moreover, the authors present an algorithmic scheme that takes any internal regret algorithm and converts it to a swap regret algorithm. This reduction is similar to that of Stoltz but attains better computational efficiency, and improves the regret in the bandit setting to \sqrt{T N^2}.

Strengths: The authors take big step towards solving a long standing open problem, with application the rate of convergence to correlated equilibrium in two-player games. The reduction that the authors present is simple and intuitive.

Weaknesses: The lower bound requires that T <= N^{3/2} (roughly) which weakens the result slightly. Moreover, I fear that the proof in the lower bound is incorrect. The proof of Lemma 2 (line 239) starts by conditioning on {T_i}, the number of time steps that the learner chooses each action. Note that, after this conditioning, the losses generated by the environment might no longer be i.i.d. Bernoulli random variables. Also, a-priori n_i - the number time steps that the learner chooses action i - are random variables. Given that, Lemmas 3 and 4, which assume that the losses are i.i.d. Bernoulli, cannot be applied conditioned on {T_i}. On the other hand, they also cannot be applied a-priori since that assume that the n_i's are fixed. The only way that I can see how to fix that is by taking a union bound on all possible choices of the n_i's which would cost an O(log T) factor in the lower bound thus making it obsolete. Can the authors please comment on that?

Correctness: See weaknesses.

Clarity: The paper is very well written, clear and easy to follow.

Relation to Prior Work: Relation to previous work is discussed extensively in Section 2 of the paper (Related Work).

Reproducibility: Yes

Additional Feedback: --- After Rebuttal --- The authors have convinced me that their proof is correct after all.


Review 2

Summary and Contributions: The paper studies online learning with swap regret, where the counterfactual is having changed each play of actions a with some other action F(a). First proves a lower bound for online learning, which tightens an existing bound by a log factor and matches the upper bound. The approach is a careful modification of the previous lower bound argument. Second, gives a reduction, using a no-external-regret algorithm to obtain a good no-swap-regret algorithm. This improves known upper bounds by log factors. The idea simplifies and tightens an approach of prior work, using a copy of the no-regret algorithm for each action and computing the action to play via a Markov chain.

Strengths: I think the problem is interesting and relevant. Although the gap in bounds was small, I think the improvement is significant. The fact that an improved upper bound is achieved via a simplification is also nice.

Weaknesses: The paper is heavily based on reference [9]. It was not always easy to follow, though I thought it was mostly well-written.

Correctness: Unfortunately I didn't have time to check the full proofs, however, the sketches seemed reasonable to me.

Clarity: Yes.

Relation to Prior Work: Yes.

Reproducibility: Yes

Additional Feedback: (none, please see summary/strengths/weaknesses) -------- After author response: Thanks for the response, I was reassured about the significance and novelty of the contribution.


Review 3

Summary and Contributions: Post-response: My opinion remains unchanged after the author response and reviewer discussions. This paper has two important contributions. First, it improves the lower bound on the swap regret of any algorithm, adding a sqrt(log(N)) factor over the best prior bound. This new bound matches the upper bound achieved by the well-known external-to-swap regret reductions, which proves the improved bound is tight. Second, this paper presents a modification to the external-to-swap regret reduction of Blum and Mansour. Though the modification itself is rather minor, its runtime is slightly better and its analysis provides an asymptotically better regret bound than the original reduction in the bandit setting.

Strengths: The theoretical claims seem sound and they are clearly significant and relevant to the NeurIPS community. The modification to the external-to-swap regret algorithm reduction is simple and easy to implement. It's possible that the analysis for the Blum and Mansour reduction can be tightened in the bandit setting, but even if that is the case the modified reduction has computational advantages.

Weaknesses: The proof of the lower bound is very similar to the previous bound from Blum and Mansour and the modification and the conditions under which it holds are slightly stronger.

Correctness: The claims appear to be correct. I checked them closely, but I'm not 100% confident in my assessment.

Clarity: Generally the paper is well written. The proofs are not as easy to follow as the original Blum and Mansour paper, but they set a high bar.

Relation to Prior Work: Yes. I would recommend adding a citation to Hart and MasCollel, Simple Adaptive Strategies: From Regret-Matching to Uncoupled Dynamics, Econometrica 2000. This paper won the SIGEcon Test of Time award this year alongside Foster and Vohra, which is cited.

Reproducibility: Yes

Additional Feedback: In table 1, doesn't Theorem 1 also improve the lowerbound for the full information setting? I don't see what in the proof restricts it to the bandit setting. I'm not sure if it's worth mentioning, but due to the external-to-swap regret reduction only updating a single external-regret minimizer on each iteration one should be able to update the singular value decomposition of the Q matrix in something like n^2*log(1/eps) time using a rank-1 update, which is something the original reduction cannot do. I think this is asymptotically the same as using the power method to solve for p, which can be done with both reductions, but it may have better conditioning.


Review 4

Summary and Contributions: This paper provides a tight lower bound for swap regret in full-information online learning. The paper adopts a classic online learning setup. There is a finite set {1, …, N} of actions. In each time step t ∈ {1, …, T}, the environment chooses a loss per action 𝓁^t = (𝓁_1^t, …, 𝓁_N^t) ∈ [0,1]^N. The player chooses an action i^t ∈ [N] (without knowledge of 𝓁^t) and incurs a loss of 𝓁^t_{i^t}. This paper studies swap regret, which compares the cumulative loss for the player and that of any swapped action sequence generated by an arbitrary modification rule F : [N] -> [N]. In other words, the swap regret is max_F{R^T(F)} = max_F sum_{t = 1}^T 𝓁^t_{i^t} - 𝓁^t_{F(i^t)}. This is a generalization of the classic external regret. In the full information setting, prior research by Blum and Mansour [’07] gave a poly-time algorithm with O(sqrt(TN * log(N))) swap regret. The authors write that the previously best-known lower bound of Ω(sqrt(TN)) was also provided by Blum and Mansour [’07]. This paper extends Blum and Mansour's ['07] lower bound technique to provide a lower bound of Ω(sqrt(TN * log(N))), which also applies to the bandit setting. They also provide a computationally efficient reduction from no-external-regret algorithms to no-swap-regret algorithms. This implies a poly-time O(sqrt(TN^2)) swap regret bound for the bandit setting, which improves upon the exponential-time O(sqrt(TN^2 * log(N))) swap regret algorithm by Stoltz [’05] and the poly-time O(sqrt(TN^3 * log(N))) swap regret algorithm by Blum and Mansour [’07].

Strengths: I believe this paper’s contribution is significant since swap regret appears to be an important generalization of external regret, with noteworthy applications in algorithmic game theory. Whether or not the O(sqrt(TN * log(N))) swap regret bound is tight appears to be a long-standing open question, and was included by Blum and Mansour in a chapter of Nisan et al.’s [’07] textbook on algorithmic game theory. While not yet tight, the poly-time bandit algorithm with O(sqrt(TN^2)) swap regret also seems to be a significant improvement over the previously best-known algorithms.

Weaknesses: While obtaining this tight guarantee is an important contribution, it’s a bit hard to tell how significant an extension this is beyond Blum and Mansour’s [’07] paper. The description of Blum and Mansour’s [’07] setup (Section 4.1) was clear and easy to follow. Here’s the basic idea from Blum and Mansour’s [’07] setup. In essence, the loss per action is a Bernoulli(1/2) random variable. However, if some action is selected too many times (>= 8T/N times), it is “blocked” in that for all future rounds, the loss is 1. Blum and Mansour [’07] prove (restated as Lemma 1 in this paper) that in order for the player to not incur Ω(T) regret, they must avoid choosing blocked actions, and thus their actions are balanced: there are Ω(N) actions that are chosen Ω(T/N) times. This fact allows Blum and Mansour [’07] to prove a Ω(sqrt(TN)) swap regret bound. I’ll highlight one particularly relevant part of the proof. Let’s say that L_{ij} is the total loss the player would have incurred if he had chosen action j on each time step that he actually choose action i. The expected swap regret can easily be bounded by T/2 - E[sum_{i = 1}^N min_{j ∈ [N]} L_{ij}]. Blum and Mansour’s [’07] proof essentially follows from bounding E[min_{j ∈ [N]} L_{ij}] for each action i ∈ [N] L_{ij}. (I’m eliding a few details here, but this is the basic idea.) From Section 4.2, it seems as though the key innovation of this paper is that instead of bounding each term E[min_{j ∈ [N]} L_{ij}] individually, the tighter bound comes from analyzing the entire sum sum_{i = 1}^N min_{j ∈ [N]} L_{ij}. (I’ll note that the authors are actually bounding a closely related sum—the right-hand-side of Equation(2)—but I won’t get into all the details here.) This doesn't quite seem like a groundbreaking modification conceptually, but since it provides the tight lower bound, that seems noteworthy enough to deserve publication. Section 4.2 was quite technical, so I would have really appreciated some more intuition. The authors write that they provide a “proof sketch,” but it’s really just a full, very technical proof of the key Lemma 2 (with the proofs of helper lemmas 3 and 4 moved to the appendix). It seemed like the authors might be providing some kind of high-level intuition about this proof in lines 87-90, but I couldn’t quite follow the connection between these two sentences, so it would be great if that could be fleshed out a bit. -----------------------After author feedback----------------------- Thanks for the clarification!

Correctness: I believe the claims are correct.

Clarity: Yes, though I pointed out a few ways the writing could be clarified in the “weaknesses” section.

Relation to Prior Work: Yes, the paper provides a very in depth discussion of the relation to prior research by Blum and Mansour [’07] and a detailed overview of other prior research in Section 2.

Reproducibility: Yes

Additional Feedback: - In lines 69-90 it was a bit hard to keep track of what was already there in Blum and Mansour’s [’07] paper and what’s new to this paper. The authors might clearly distinguish this with two separate paragraphs. - Line 237: There’s an extra <= T/2 inside of the expectation.

[Author Response · NeurIPS 2020]

**Dear Reviewer#1:**

> The lower bound requires that $T <= N^{3/2}$ (roughly) which weakens the result slightly.

We agree with this comment. As we mentioned in lines 63–68, we believe that this requirement can be removed by more sophisticated analysis. This requirement comes from the fact that our proof relies on a union bound on all possible choices of actions, which is closely related to the following concern and questions posted by the reviewer.

> The proof of Lemma 2 (line 239) starts by conditioning on $\{T_i\}$, ...

Lemma 2 is **not** about the posterior distribution conditioned on $\{\mathcal{T}_i\}_{i=1}^N$, but we regard the value $\sum_{i=1}^N \min_{j \in [N] \setminus B} L'_{ij}$ as a function in $\{\ell'_t\}_{t=1}^T$ for an arbitrarily fixed $\{\mathcal{T}_i\}_{i=1}^N$. Hence, we may assume the losses are i.i.d. Bernoulli. We shall stress this in the revised version.

> The only way that I can see how to fix that is by taking a union bound on all possible choices of the $n_i$'s

Yes, as the reviewer mentions, we indeed applied a union bound on all possible choices of $\{\mathcal{T}_i\}$'s, which is explained in lines 230–234. (We applied union bound because posterior distributions are not i.i.d. Bernoulli as the reviewer pointed out, which seem too complicated to analyze.) The union bound leads to $\Omega(\sqrt{TN \log N})$-lower bound (though the assumption of $T = \Omega(N^{3/2}/\log N)$ is required here) as follows: The number of possible choices is at most $N^T = \exp(T \log N)$ and, from Lemma 2, the probability that $\sum_{i=1}^N \min_{j \in [N] \setminus B} L'_{ij} > \frac{T}{2} - \frac{\sqrt{TN \log N}}{512}$ is at most $2 \exp(-N^{3/2}/128)$ for each choice of $\{\mathcal{T}_i\}$. Hence, $\sum_{i=1}^N \min_{j \in [N] \setminus B} \leq \frac{T}{2} - \frac{\sqrt{TN \log N}}{512}$ for all possible choices with probability at least $1 - 2 \exp(-N^{3/2}/128 + T \log N)$. This probability is $\Omega(1)$ under the assumption of $T = O(N^{3/2}/\log N)$, which leads to an $\Omega(\sqrt{TN \log N})$-lower bound for swap regret, as discussed in lines 228–238.

**Dear Reviewer#2:**

> The paper is heavily based on reference [9].

Yes, the reference [9] is the most important previous work the present paper relies on. For the key update presented in our study, we would like you to refer to the comment by Reviewer#4 and our response to it.

**Dear Reviewer#3:**

> I would recommend adding a citation to Hart and MasCollel, ...

Thanks for providing information on relevant work. The revised manuscript shall refer to this paper.

> In table 1, doesn't Theorem 1 also improve the lowerbound for the full information setting?

As stated in the caption of Table 1, the lower bound in Theorem 1 applies to the full-information as well as to the bandit settings.

**Dear Reviewer#4:**

> From Section 4.2, it seems as though the key innovation of this paper is that instead of bounding each term $E[\min_{j \in [N]} L_{ij}]$ individually, the tighter bound comes from analyzing the entire $\sum_{i=1}^N \min_{j \in [N]} L_{ij}$. ..

Yes, the point the reviewer mentioned is a key idea of our analysis. An intuition for an additional $\Omega(\sqrt{\log N})$ factor comes from a property of order statistics: for $k = O(N)$, the $k$-th largest value of $\{n_i/2 - L'_{ij}\}_{j \in [N]}$ is $\Omega(\sqrt{n_i \log(N/k)})$ with high probability (as can be seen from Lemma 10). Combining this and the fact that $L_{ij} = L'_{ij}$ holds for any non-blocked action $j$, we want to get a tighter upper bound for $\min_{j \in [N]} L_{ij}$. When bounding $\min_{j \in [N]} L_{ij}$ individually as the previous work, since the number of blocked action is at most $(N/8)$, by considering the *worst-case* w.r.t. the blocked actions (when top $N/8$ actions regarding $\{n_i/2 - L'_{ij}\}_{j \in [N]}$ are blocked) we have $\max_{j \in [N]}\{n_i/2 - L_{ij}\} = \Omega(\sqrt{n_i \log(N/(N/8))}) = \Omega(\sqrt{n_i})$. This bound is not satisfactory since there is no $\log N$-factor. On the other hand, when analyzing the entire $\sum_{i=1}^N \min_{j \in [N]} L_{ij}$ as in our study, we can exploit the fact that the blocked actions are *shared by all $i$'s*, to improve upon the worst-case analysis w.r.t. blocked actions. As formally stated in Lemma 4, we can see that top $\sqrt{N}$ actions ($j$'s) cannot be blocked for most of $i$'s. For such $i$'s, we get $\max_{j \in [N]}\{n_i/2 - L_{ij}\} = \Omega(\sqrt{n_i \log(N/\sqrt{N})}) = \Omega(\sqrt{n_i \log N})$, which includes an additional $\Omega(\sqrt{\log N})$ factor.

> what was already there in Blum and Mansour 's [ ' 07] paper and what 's new to this paper.

Thanks for your helpful comments. In the revised version, we shall divide the paragraph into the parts of existing ideas and new ones, in order to clarify what's new.

[Meta-Review · NeurIPS 2020]

All reviewers support the paper. Clear accept. Please do improve the clarity of the paper as promised in the rebuttal.